# Kraken: Inherently Parallel Transformers For Efficient Multi-Device Inference

**Rohan Baskar Prabhakar**
Princeton University
rohanbp@princeton.edu

**Hengrui Zhang**
Princeton University
hengrui.zhang@princeton.edu

**David Wentzlaff**
Princeton University
wentzlaf@princeton.edu

## Abstract

Large Transformer networks are increasingly used in settings where low inference latency is necessary to enable new applications and improve the end-user experience. However, autoregressive inference is resource intensive and requires parallelism for efficiency. Parallelism introduces collective communication that is both expensive and represents a phase when hardware resources are underutilized. Towards mitigating this, Kraken is an evolution of the standard Transformer architecture that is designed to complement existing tensor parallelism schemes for efficient inference on multi-device systems. By introducing a fixed degree of intra-layer model parallelism, the architecture allows collective operations to be overlapped with compute, decreasing latency and increasing hardware utilization. When trained on OpenWebText, Kraken models reach a similar perplexity as standard Transformers while also preserving their language modeling capabilities as evaluated on the SuperGLUE benchmark. Importantly, when tested on multi-GPU systems using TensorRT-LLM engines, Kraken speeds up Time To First Token by a mean of $35.6\%$ across a range of model sizes, context lengths, and degrees of tensor parallelism.

## 1   Introduction

Deep neural networks based on the Transformer architecture (51) have become the prevalent choice for a variety of tasks involving sequences, especially in natural language processing and computer vision (44) (7) (19). Their capabilities, particularly in language modeling, have been driven by a rapid increase in parameter count (7). Today's largest language models have up to a trillion parameters (21) and consequently demand more efficiency from the systems used to train and serve them. This has necessitated the need for many techniques and optimizations that focus on improving the performance of both algorithms and systems (47) (34) (17) (37) (12).

Large models are often used in interactive applications where latency is an important metric that dictates the quality of the end-user experience (28). A typical web search takes about $0.2$ seconds but the Time To First Token (TTFT) for large models can be up to a few seconds (depending on context length, model size, and available hardware) (42). Additionally, because it is not always feasible to run models on local hardware, they are served to users via datacenters that use multi-device compute nodes, adding to latency constraints. Increasingly, language models are also used as intermediate steps in longer processes such as augmenting web searches or presenting the results of database queries (28). The rising prevalence of such multi-step applications makes reducing inference latency even more critical.

38th Conference on Neural Information Processing Systems (NeurIPS 2024).

Continuing this theme, this work focuses on reducing the latency cost of the collective operations introduced by tensor parallelism (49) in the forward pass. In particular, it introduces Kraken, a variation of the standard Transformer architecture (44)(7) that reduces the amount of inter-device communication and allows remaining collective operators to be overlapped with compute. Kraken models have a fixed degree of innate model parallelism that allows computational graphs on each device to run independently without having to wait for the results of collective operations. The architecture is designed to complement the topology of multi-device setups such as nodes in typical datacenters and DGX (13) systems. By designing the model architecture to account for characteristics of the hardware, our approach increases compute utilization and allows more efficient inference.

We evaluate the improvements Kraken offers over standard Transformers in two key aspects: model quality and inference latency. For the former, we train a series of Kraken models with varying degrees of parallelism and parameter count on OpenWebText (23) and compare them with the GPT-2 (44) family of models on the SuperGLUE suite of benchmarks (53). We then implement Kraken using the TensorRT-LLM library (15) and measure the Time To First Token (TTFT) given various model sizes and context lengths to illustrate the efficiency gains when collective operators are no longer on the critical path. We find that while maintaining the language modeling capabilities of standard Transformers, Kraken models speedup the Time To First Token (TTFT) by a geomean of $35.6\%$ when tested across a range of model sizes, context lengths, and degrees of parallelism.

## 2 Background

### 2.1 Decoder-Only Transformer models

We will briefly discuss the forward pass of decoder-only Transformer (DTransformer) models that use self-attention mechanisms to perform language modeling (39) in order to motivate our approach. Given an input sequence $x$ consisting of tokens belonging to a vocabulary $V$, such models return a probability distribution over the vocabulary that describes what the next token in $x$ could be i.e., the model is trained to estimate $P(x[\ell + 1] \mid x[1 : \ell])$ where $\ell$ is the initial length of $x$. To compute the output logits in the forward pass, $x$ is converted to a sequence of embeddings that incorporate information about each token and its position in the sequence. These embeddings are used as input to a stack of Transformer layers one of which is depicted in Figure 1.

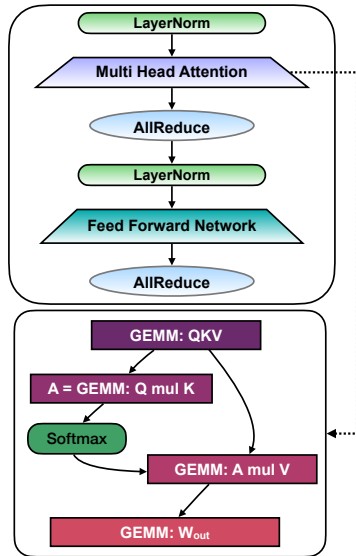

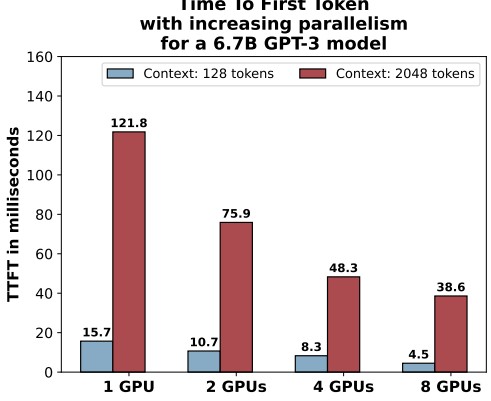

Figure 2: **Increasing the degree of tensor parallelism decreases the Time To First Token.** Even when weights and KV cache fit on device memory, parallelism can be worthwhile. These results are for a 6.7B parameter GPT-3 like model and were collected using TensorRT-LLM engines on our evaluation platform: an HGX A100 40GB system.

Figure 1: **One layer of a standard Transformer** consisting of Multi-Head Attention (also shown) followed by a FeedForward Network. Residual connections have been omitted.

Such layers comprise the bulk of the compute in the model and consist of a Multi-Head Attention (MHA) block followed by a FeedForward Network (FFN or Multi-Layer Perceptron). The FFN typically consists of two linear transformations with a non-linear activation function in between. On the other hand, Multi-Head Attention (Figure 1) implements scaled dot-product attention i.e., each head computes $\texttt{Softmax}(\frac{QK^T}{\sqrt{h}})V$ where $Q, K, V \in \mathbb{R}^{h \times l}$. Here, $h$ is the head dimension and $l$ is the sequence length. The General Matrix Multiply (GEMM) $W_{out}$ combines the outputs of the different heads. The activations of the last layer are used as input to the unembedding operation. Henceforth, we will refer to the GPT-2,3 (44)(7) like construction as the standard Transformer architecture. This variant uses MHA, has sequential Attention and FFN blocks, expands the embedding dimension from $d$ to $4d$ in the hidden layer of the FFN, uses Gaussian Error Linear Unit (GELU) non-linearities, and places Layer Norm operators before the MHA and FFN i.e., is a Pre-LN Transformer.

A widely used optimization during inference involves caching the Key and Value matrices (from GEMM $QKV$ in Figure 1) of each token in a KV cache that is stored in memory. This memoization has the effect of breaking up autoregressive inference (38)(58) into two distinct steps: 1) Prefill (when the first token and the KV cache are generated) and 2) Decode(for all subsequent tokens). Both these steps have distinct runtime characteristics with Prefill being more compute bound and Decode being more memory bandwidth bound (58). The KV cache entry for the next generated token is appended during each Decode step. Prefill, measured by TTFT, typically takes much longer than a single Decode step and will be the focus of this work.

## 2.2 Multi-Device systems

Given the extensive amount of compute and memory capacity required to efficiently serve large models, most widely used systems are node-based configurations where each node has a small number (between 4 and 16) of devices. These devices usually take the form of Graphics Processing Units (GPUs) or Tensor Processing Units (TPUs). The discussion in this work will focus on GPUs but we expect that our findings will also be of relevance to other choices of accelerators such as TPUs. Accessing data at levels of the system physically closer to compute cores such as scratchpad memory or caches is typically much faster and more efficient compared to accessing off-chip memory. Techniques like tiled matrix multiplication (20) and FlashAttention (17) account for this characteristic, considerably improving the runtime performance of implementations.

Devices within a node are configured in a topology and linked by interconnects such as Peripheral Component Interconnect Express (PCIe), NVLink, and NVSwitch. The different standards balance versatility and cost with performance. For instance, a topology that uses PCIe switches across some connections will have less overall bandwidth than a system that uses solely NVLink/NVSwitch. Inter-device communication primitives are provided to other software by libraries like NCCL (14) and RCCL (11). Communication is comparatively expensive and represents a phase in the forward pass where compute cores are mostly idle. This work strives to extend the IO-aware approach used by techniques like FlashAttention (17) towards the multi-device setting.

## 2.3 Tensor parallelism

Serving large models in multi-device settings requires parallelization schemes and strategies that partition the input sequence, model weights, activations, and/or incoming inference requests (49) (56) (35) (1) (60). In particular, model parallel (intra-operator) schemes fall into two broad categories: tensor parallelism and pipeline parallelism. Since Kraken models are designed to improve inference latency, this discussion will center around tensor parallelism.

There are several possible strategies (42) (55) to achieve distributed tensor parallelism in Transformer models but we will focus on the widely used scheme introduced by Shoeybi et al. (49) which is well-suited for multi-GPU settings. This scheme introduces two AllReduce operations per layer and takes advantage of the implicitly parallel nature of Multi-Head Attention with optimal partitioning of the FeedForward Network. In each layer, contiguous groups of Attention heads are placed across different devices and the $W_{out}$ matrix that is used to combine the output of the different heads is partitioned across columns. The output of the MHA block is retrieved by reducing the local output of all devices; this introduces one AllReduce as shown in Figure 1. Similarly, in the FFN block, the $W_1$ weight matrix is partitioned across rows (using the notation where $W_1$ is multiplied by a column-vector of activations) and the $W_2$ weight matrix is partitioned across columns. The output

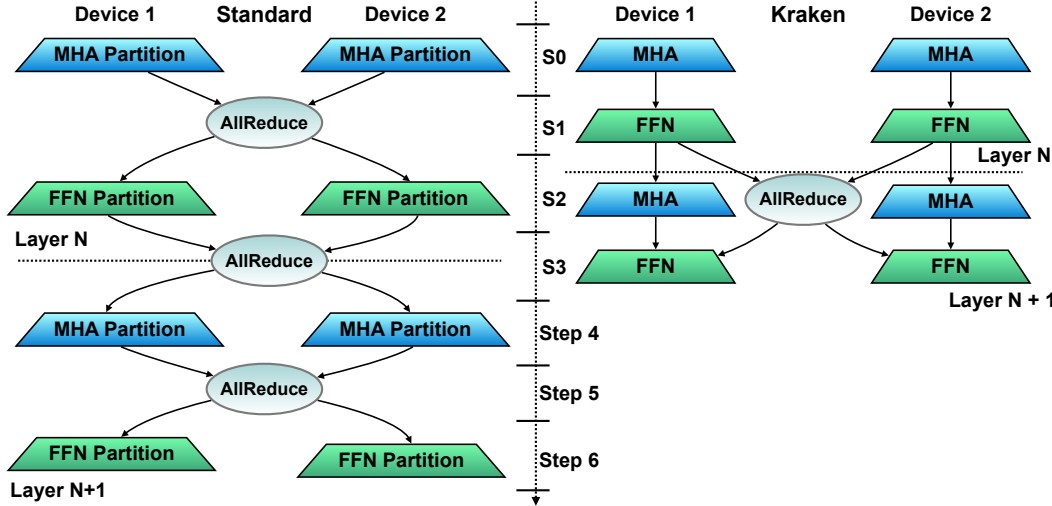

Figure 3: **Parallelizing two standard Transformer layers compared to executing two layers of a Kraken Transformer with 2-way parallelism**. Kraken Transformers have fewer AllReduce operations and these can be run concurrently with the Multi-Head Attention in the next layer. Step lengths are illustrative and not indicative of how much wall-clock time a particular operation might require.

of the FFN is computed by reducing the local output of all devices, thereby introducing another AllReduce operation. The objective of this work is to reduce and hide the runtime impact of these collectives introduced by tensor parallelism.

# 3 Kraken: Model architecture

## 3.1 Layer construction

The primary objective of Kraken is to preserve as much of the standard Transformer (GPT-2,3 like)(44)(7) architecture as possible while reducing the latency cost of the collective communication introduced by tensor parallelism. To achieve this, we allow each of the individual shards of a parallelized Transformer layer to behave as independent, smaller layers. More precisely, as depicted in Figure 3, instead of sharding the Multi-Head Attention and FeedForward Network blocks, each MHA and FFN block is replaced with a smaller, independent block. This introduces *a fixed degree of parallelism* that is chosen at the outset of training and accounts for the most common hardware deployment target. For example, if a model will be mostly served on nodes with eight GPUs each, a reasonable choice would be to use either 4-way or 8-way parallelism (depending on the size of the model). The former would be suitable for smaller models, allowing each node to serve two different inference requests at the same time.

Consequently, the only inter-device dependency is an AllReduce operation at the end of every layer. This collective represents the only interchange of activations between groups of sub-layers. Its output is used in the FFN block of the next layer and not in the MHA block. As shown in Figure 3, this allows for overlapping the compute in the MHA block with the AllReduce, effectively removing most inter-device communication from the critical path. Much like how positional embeddings are added to the token embeddings prior to the first layer (39), we chose to use element-wise addition to combine the outputs of the various sub-layers from the previous layer. This occurs prior to the Layer Norm before the FFN. The forward pass of each sub-layer, including residual connections, is also described in Algorithm 1. All pretrained models in our evaluation use the GELU activation function in the FFN. In initial experiments that explored different constructions, we scaled the weights of the residual layer by $\frac{1}{\sqrt{L*N}}$ using a similar line of reasoning as that used by Radford et al. (44). Here, $L$ is the number of layers, and $N$ is the degree of parallelism. We did not remove this initialization scheme in subsequent experiments even though accumulation along residual connections was limited to groups of $L$ sub-layers.

**Algorithm 1:** Kraken Sub-Layer: Forward Pass

**Input:** $x \in \mathbb{R}^{l \times d}$
**Output:** $y \in \mathbb{R}^{l \times d}$

1   $residual = x$

    /* The first layer replaces the AllReduce with the identity operator            */

2   $y = \texttt{AllReduce}(x)$

3   $x = \texttt{LayerNorm}(x)$

4   $x = residual + \texttt{MultiHeadAttention}(x)$

5   $residual = x$

    /* The output of the AllReduce is used only here allowing it to be overlapped
      with Attention                                                   */

6   $x = \texttt{LayerNorm}(x + y)$

7   $y = residual + \texttt{FeedForwardNetwork}(x)$

8   return $y$

---

The token and positional embeddings are shared across all sub-layers of the first layer i.e., there is still one set of embeddings to maintain compatibility with weight tying (27) (43). After the last layer, we combine the outputs of the different sub-layers using a linear transformation with weights $W_{concat} \in \mathbb{R}^{d*N \times d}$ where $d$ is the embedding dimension and $N$ is the degree of parallelism. The output of this transformation is used as input to the unembedding. Depending on the implementation, this linear layer introduces the only blocking collective in the computational graph.

### 3.2   Deriving model configurations for a fixed parameter budget

Increasing the degree of parallelism while keeping other hyperparameters like the embedding dimension constant will increase the parameter count. Instead, using a configuration of a standard Transformer as the basis, we make the following two modifications to derive a Kraken configuration that has approximately the same number of parameters:

- First, the hidden state expansion in the FFN is reduced from $4d$ to $2d$ where $d$ is the embedding/model dimension.

- Given the number of parameters $P$, degree of parallelism $N$, number of layers $L$, and vocabulary size $V$, we derive a closed form expression for $P$ and solve for $d$ i.e., $P = V*d + \Sigma_{i=1}^{L} N * ((3*(d \times d) + (d \times d)) + 2*(d \times 2d))$ where the term $((3*(d \times d) + (d \times d))$ comes from MHA and $2*(d \times 2d))$ comes from the FFN.

## 4   Evaluation

### 4.1   Model configurations and perplexity

To evaluate language modeling performance, we train a series of models up to 761 million parameters large and with varying degrees of parallelism on OpenWebText (23). This allows us to compare the performance of the Kraken architecture with the GPT-2 (44) family of models. Because of limited access to compute, we do not exhaustively search for hyperparameters and stop training at 150 billion tokens in contrast to the about 300 billion tokens that language models of such sizes are typically trained for (44)(24). Table 1 details the embedding dimensions, number of layers, and other hyperparameters for each configuration. For Kraken configurations, the number of Attention heads in each layer is summed across all sub-layers. The context length was set at 1024 tokens. Models similar in size were trained using the same learning rate schedule and for the same number of gradient steps. More details about the training setup, required compute, and how each configuration was derived can be found in Appendix A.1. Pertinent code including the TensorRT-LLM implementation is available at `https://github.com/rohan-bp/kraken`.

Perplexity measurements for GPT-2 models are provided to add context to the results with the caveat that all Kraken models were trained on OpenWebText but GPT-2 models were trained on the closed-source WebText dataset. We would expect that all things equal, Kraken models will have lower perplexity unless the GPT-2 models were subsequently fine-tuned on OpenWebText. Nonetheless,

Table 1: **Model configurations and perplexity on OpenWebText for Kraken models compared to similarly sized GPT-2 models.** Lower perplexity is better.

| Model | Layers | Embedding Dimension | Attention Heads | Total Params. | Validation Perplexity |
|---|---|---|---|---|---|
| GPT-2 | 12 | 768 | 12 | 117M | 20.64 |
| Kraken 2-way | 12 | 678 | 12 | 124M | 18.89 |
| Kraken 4-way | 12 | 504 | 12 | 124.5M | 18.56 |
| Kraken 6-way | 12 | 418 | 12 | 123.2M | 19.22 |
| GPT-2 Medium | 24 | 1024 | 16 | 345M | 14.87 |
| Kraken 2-way | 24 | 888 | 16 | 350M | 14.40 |
| Kraken 4-way | 24 | 644 | 16 | 353.4M | 14.71 |
| GPT-2 Large | 24 | 1280 | 20 | 762M | 13.69 |
| Kraken 4-way | 24 | 960 | 16 | 761M | 13.09 |

Table 2: **Zero-Shot performance on SuperGLUE.** ReCoRD uses the F1 score as the evaluation metric. All other benchmarks use accuracy.

| Model | BoolQ | RTE | CB | COPA | ReCoRD | WIC | WSC | MultiRC | Average |
|---|---|---|---|---|---|---|---|---|---|
| GPT-2 | 48.38 | 51.99 | 41.07 | 62.0 | 71.07 | 49.53 | 43.27 | 53.47 | 52.6 |
| Kraken 124M (2-way) | 53.85 | 54.15 | 44.64 | 68.0 | 72.42 | 49.53 | 36.54 | 56.89 | 54.5 |
| Kraken 124M (4-way) | 50.15 | 53.79 | 41.07 | 67.0 | 71.47 | 50.31 | 37.5 | 57.16 | 53.56 |
| Kraken 124M (6-way) | 47.92 | 56.68 | 8.93 | 69.0 | 70.69 | 50.16 | 53.85 | 53.82 | 51.38 |
| GPT-2 Medium | 58.53 | 53.07 | 42.86 | 68.0 | 79.43 | 50.0 | 41.35 | 52.58 | 55.73 |
| Kraken 355M (2-way) | 60.06 | 51.62 | 41.07 | 69.0 | 80.04 | 50.0 | 35.58 | 56.91 | 55.54 |
| Kraken 355M (4-way) | 61.68 | 55.23 | 35.71 | 72.0 | 79.01 | 50.0 | 36.54 | 57.2 | 55.92 |
| GPT-2 Large | 60.55 | 52.71 | 41.07 | 72.0 | 81.95 | 49.69 | 44.23 | 48.56 | 56.34 |
| Kraken 760M (4-way) | 60.58 | 49.1 | 10.71 | 73.0 | 82.04 | 50.0 | 36.54 | 51.65 | 51.7 |

even when trained over a smaller number of tokens, Kraken models reach a similarly low perplexity as standard Transformers.

## 4.2 Performance On SuperGLUE

We used the SuperGLUE benchmark suite (53) to evaluate performance on language tasks. All benchmarks were scored on accuracy except for ReCoRD which uses the F1 Score instead. No finetuning or training was performed for any combination of model and benchmark. Table 2 contains results for Zero-Shot performance and Table 3 presents performance in the Three-Shot setting. Scores were calculated using the Language Model Evaluation Harness (2) with the default choice of prompts and scoring metrics for option `lm-eval-SuperGLUE v1`. As conveyed by these results, Kraken largely preserves the language modeling capabilities of the standard Transformer architecture. We expect that the gap between standard Transformers on language tasks will close further if we train the models on higher quality data and using optimal choices for the various hyperparameters.

## 4.3 Evaluation platform

To measure improvements in inference latency, we used the TensorRT-LLM (15) library to create engines and compare Kraken models with other widely used dense model architectures. The library provides an interface to define popular Transformer models and bundles a collection of kernels,

Table 3: **Three-Shot performance on SuperGLUE.** ReCoRD uses the F1 score as the evaluation metric. All other benchmarks use accuracy.

| Model | BoolQ | RTE | CB | COPA | ReCoRD | WIC | WSC | MultiRC | Average |
|---|---|---|---|---|---|---|---|---|---|
| GPT-2 | 53.76 | 47.65 | 42.86 | 60.0 | 70.12 | 50.0 | 51.92 | 50.62 | 53.37 |
| Kraken 124M (2-way) | 55.14 | 48.74 | 44.64 | 61.0 | 68.83 | 49.69 | 47.12 | 50.56 | 53.22 |
| Kraken 124M (4-way) | 59.05 | 44.04 | 48.21 | 64.0 | 69.94 | 47.49 | 51.92 | 54.17 | 54.85 |
| Kraken 124M (6-way) | 54.28 | 46.93 | 42.86 | 63.0 | 69.17 | 49.37 | 44.23 | 51.96 | 52.72 |
| GPT-2 Medium | 60.58 | 47.65 | 44.64 | 68.0 | 78.35 | 47.65 | 50.0 | 53.22 | 56.26 |
| Kraken 355M (2-way) | 50.73 | 53.07 | 37.5 | 68.0 | 79.01 | 49.06 | 66.35 | 51.34 | 56.88 |
| Kraken 355M (4-way) | 61.8 | 51.99 | 50.0 | 73.0 | 77.97 | 47.49 | 44.23 | 54.17 | 57.58 |
| GPT-2 Large | 60.58 | 51.99 | 39.29 | 70.0 | 81.05 | 47.81 | 60.58 | 51.53 | 57.85 |
| Kraken 760M (4-way) | 55.75 | 53.79 | 42.86 | 68.0 | 81.03 | 45.77 | 62.5 | 53.47 | 57.9 |

Table 4: **Configurations for the different model engines used to compare TTFT.** For models of similar sizes, hyperparameters are shared for the GPT-like and Parallel Attention + FeedForward variants. 4-way denotes Kraken configurations used when evaluating tensor parallelism across 4 devices and likewise for 8-way.

| Model Size | Layers | dModel | Params. Per Layer | Attention Heads | Kraken 4-way dModel | 4-way Params. Per Layer | 4-way Attention Heads | Kraken 8-way dModel | 8-way Params. Per Layer | 8-way Attention Heads |
|---|---|---|---|---|---|---|---|---|---|---|
| 1.3B | 24 | 2048 | 50.3M | 16 | 1248 | 49.9M | 12 | 960 | 59.0M | 10 |
| 6.7B | 32 | 4096 | 201.3M | 32 | 2496 | 199.4M | 24 | 1920 | 235.9M | 20 |
| 13B | 40 | 5140 | 317.0M | 40 | 3120 | 311.5M | 30 | 2304 | 339.7M | 32 |
| 65B | 80 | 8192 | 805.3M | 64 | 4992 | 797.4M | 39 | 3648 | 851.7M | 38 |
| 175B | 96 | 12288 | 1.81B | 96 | 7424 | 1.76B | 58 | 5472 | 1.92B | 57 |

plugins, and other optimizations that can be used to create efficient TensorRT engines (containing model weights) and serve them on systems with GPUs. All experiments were conducted on a 8 x A100 GPU machine with NVSwitch and 40GB of HBM memory per GPU.

### 4.4 Speedup in Time To First Token

For comparisons with other model architectures, we build engines for standard, GPT-like configurations and GPT-J (54) like configurations as detailed in Table 4. The latter serves to contrast our approach with one that runs the FFN in parallel with the Attention (10) (54). Such parallel layers require only one AllReduce in a layer but unlike Kraken, the collective cannot be overlapped with compute. For similarly sized models, the only difference is the embedding dimension for the Kraken models. All configurations follow the convention where dModel is divided by the number of Attention heads to calculate the size of each head. Other configuration parameters such as the number of layers, maximum context length, and vocabulary size were the same. The Attention heads are per sub-layer for Kraken models and parameter counts do not include biases and layer normalization.

There are two sets of engines, one for 4-way tensor parallelism and another for 8-way parallelism. We also did not follow the earlier convention 3.2 of solving for the embedding dimension of a Kraken model precisely. This is because the available optimizations in TensorRT-LLM are compatible only with specific Attention head dimensions. To account for this, we handpicked embedding dimensions that, given an equivalent GPT-like configuration, have about the same number of parameters while still being compatible with the available kernels. This was necessary for a fair evaluation but in practice, the chosen embedding dimension should also account for the performance

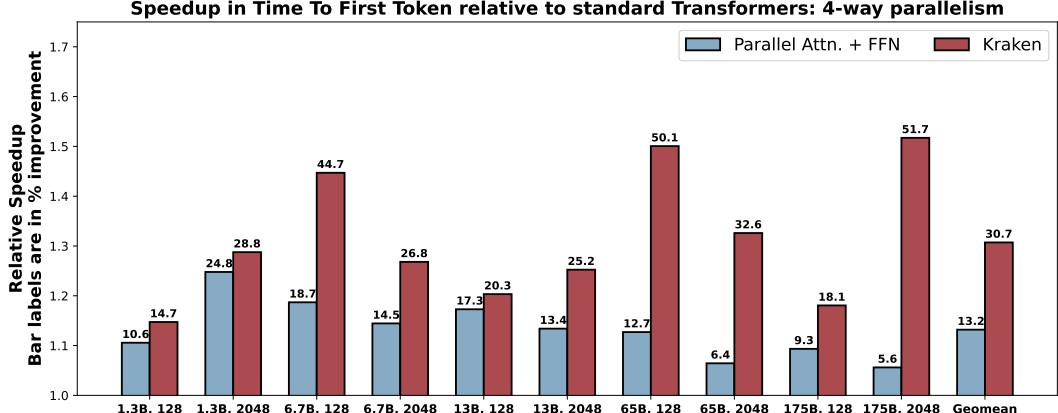

Figure 4: **Speedup in Time To First Token over standard Transformers on a system that uses NVSwitch and with 4-way parallelism**. x-axis labels denote the size of the model followed by the context length. Bar labels are in percentage improvement.

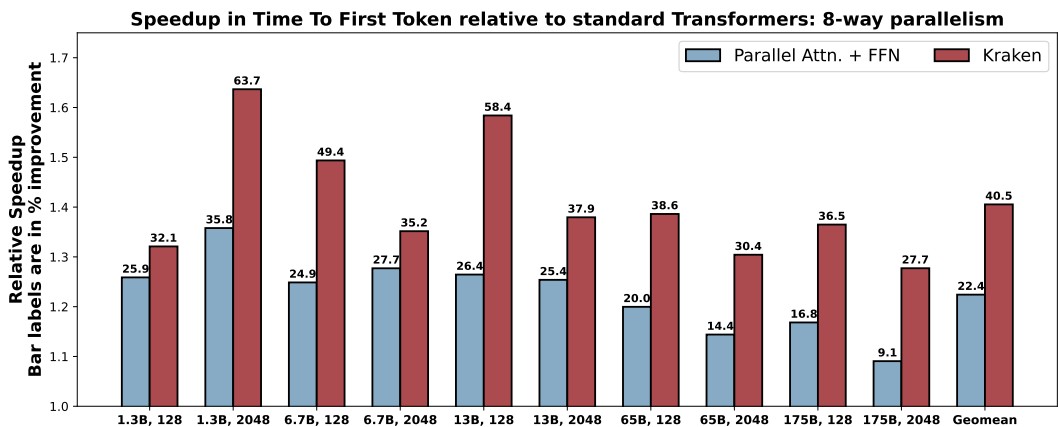

Figure 5: **Speedup in Time To First Token over standard Transformers on a system that uses NVSwitch and with 8-way parallelism**. x-axis labels denote the size of the model followed by the context length. Bar labels are in percentage improvement.

of the computational kernels it would map to. Anthony et al. (4) discuss this, showing how model hyperparameters can affect GEMM performance and consequently training efficiency.

Figure 4 shows the speedup in latency compared to GPT-like models on our evaluation platform and with 4-way parallelism while Figure 5 depicts improvements for 8-way parallelism. Across a range of model sizes and context lengths of 128 and 2048 tokens, using Kraken models can improve inference latency by anywhere from 10.9% to 63.7%. These results are end-to-end and include the extra fully-connected layer required by Kraken models. All results are normalized to the equivalent latency for a similar GPT-3 like model and engines were generated with random weights and *fp16* precision. Latency in terms of milliseconds is reported in Appendix A.3. Importantly, our evaluation platform uses NVSwitch and consequently has considerable inter-device bandwidth (600 GB/s) which means communication can be relatively inexpensive. We expect that these gains will be more pronounced on systems with less performant interconnects.

A more detailed evaluation on precisely which operators contribute to overall runtime is presented in Section 4.5. In order to take advantage of the concurrent communication and compute that Kraken allows, we extended TensorRT-LLM with the Overlap plugin that is described in Appendix A.2.

Each configuration was evaluated using the *gptSessionBenchmark* that is available as part of TensorRT-LLM. We enabled the default set of optimizations which include GPT Attention plugin, Remove Input Padding, and GEMM plugin. Experiments were run with a batch size of 1 but engines were

built with a maximum batch size of $4$, vocabulary size of $51,200$ tokens, maximum input context length of $2048$, and maximum output length of $4096$.

## 4.5 Runtime characterization

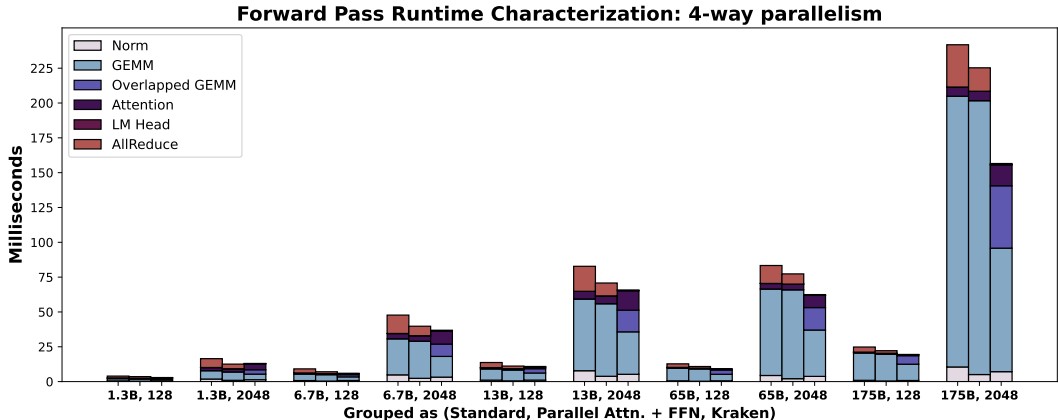

Figure 6: **Runtime characterization: 4-way parallelism.** For each cluster on the x-axis, labels denote the size of model followed by the context length.

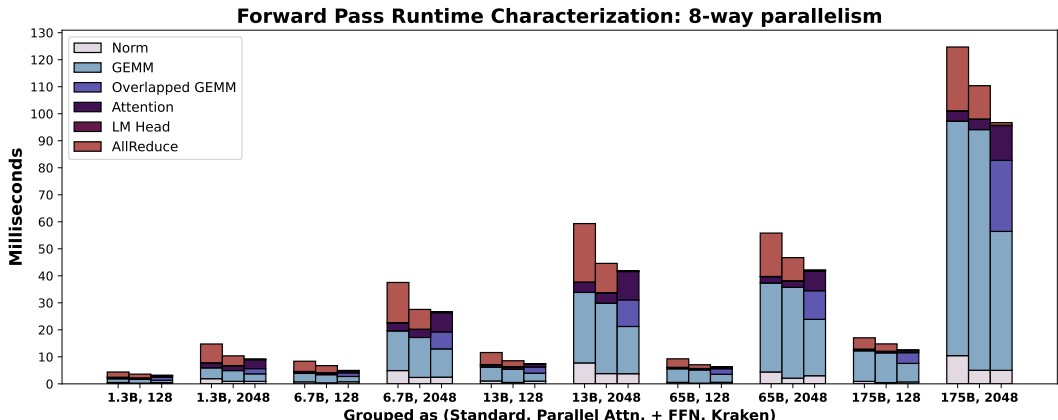

Figure 7: **Runtime characterization: 8-way parallelism.** For each cluster on the x-axis, labels denote the size of model followed by the context length.

To provide more context to the performance gains presented in Figures 4 and 5, we characterize the entire forward pass using the same experimental setting as Section 4.4. The runtime profiles presented in Figures 6 and 7 were obtained using the profiler built into *gptSessionBenchmark*. Across all model sizes and both context lengths, we find that a significant proportion of time is spent in the `AllReduce`. As expected, this proportion increases for 8-way parallelism compared to 4-way parallelism since each device works on a smaller fraction of the compute.

In some cases, such as for 175B models at context length 2048, some of the performance gains come from the GEMMs requiring significantly more time for the GPT and GPT-J like configurations. Despite these performance anomalies, in general, Kraken models spend a much smaller fraction of runtime in inter-device communication leading to considerable performance and efficiency improvements. Note that the cost of the memory copies and synchronization necessary for the Overlap plugin (Appendix A.2) are implicitly included in the `Overlapped GEMM` fraction. More precisely, `Overlapped GEMM` is the time spent computing GEMM $W_{out}$, GEMM $QKV$, and other operations in the Overlap plugin.

# 5 Discussion and Limitations

**Need for pretraining:** Currently, our approach requires training Kraken models from scratch which can be very resource intensive. Consequently, developing techniques to distill learned weights from existing models possibly as part of the weight initialization scheme is a promising area of future work. We also do not compare with more state-of-the-art Transformer training recipes or alternatives because of limited access to GPU compute. Training larger, compute-optimal (25) Kraken models on higher-quality datasets will permit evaluations on newer, more complex language modeling benchmarks. The fixed degree of model parallelism also places restrictions on the optimal choice of hardware to run a model on. For example, if we tried to deploy a model with 4-way parallelism on a system with two or six devices, we would either need to run groups of two sub-layers together or introduce more collectives. Either approach might negate most of the latency gains offered in the first place.

**Compatibility with standard Transformer improvements:** Since Kraken models replicate the Attention block by the degree of tensor parallelism, the size of the KV cache will also be larger than that of an equivalent Transformer model. One way to mitigate the increased memory consumption would be to replace Multi-Head Attention with either Multi-Query Attention (47) or Grouped Query Attention (3). Similarly, many other improvements to the standard Transformer architecture are also applicable to Kraken models such as RoPE embeddings (50) and RMSNorm (57). The architecture is also compatible with "drop-in" replacements for Attention that are more efficient because of time-complexity and/or sparsity (31) (9). Furthermore, we expect that the notion of parallelizing individual layers of the model will also prove useful in large deep learning models that use constructions other than Attention or MLPs (24). Incorporating a communication-aware approach to Neural Architecture Search (8) for Transformers is another promising area of future work.

**Applicability of existing optimizations:** We expect that Kraken will be readily compatible with many existing system-aware techniques used in deploying standard Transformers (56) (32) (1) (35). For example, it provides an extra degree of freedom to the various partitioning strategies proposed by Pope et al. (42). It is also compatible with techniques like FlashAttention (17), Speculative Decoding (34), fusing Attention with the FFN (36), and PagedAttention (32).

**Mixture-of-Experts and hybrid models:** By virtue of its architecture, Kraken evokes a comparison to Mixture-of-Experts (MoE) architectures such as the SwitchTransformer (21)(48). Layers in current MoE models have a single Attention block and use a learned Router to direct each token to one of a set of FeedForward Networks. Because the forward pass activates only a fixed fraction of the parameters, MoE models can be much larger than dense Transformers while maintaining the computational profile of inference. Nonetheless, serving them efficiently in multi-device settings is a challenge: MoEs require dynamic routing and suffer from load balancing issues. For instance, Huang et al. (26) find that the required All-to-All collectives can comprise a significant fraction of inference latency. An interesting direction of future work would be to incorporate an inter-device IO-aware approach in the construction of MoE models. In addition, recent work (52) has found that hybrid architectures that combine Multi-Head Attention, State Space Models (SSMs) (24), and FFNs in the same model perform better than models that use solely SSMs or MHA for the sequence-to-sequence transformation. Similarly, Kraken layers can be combined with standard Transformer layers and pure SSM layers to develop high-quality language models that are also efficient to run on hardware.

# 6 Acknowledgements

This material is based on research sponsored by the Air Force Research Laboratory (AFRL) and the Defense Advanced Research Projects Agency (DARPA) under agreement No. FA8650-18-2-7862. The U.S. Government is authorized to reproduce and distribute reprints for Governmental purposes notwithstanding any copyright notation thereon. The views and conclusions contained herein are those of the authors and should not be interpreted as necessarily representing the official policies or endorsements, either expressed or implied, of the Air Force Research Laboratory (AFRL), the Defense Advanced Research Projects Agency (DARPA), or the U.S. Government. We thank Princeton Research Computing for their assistance in setting up and maintaining the necessary compute infrastructure. We thank Ashwinee Panda for feedback and aid in brainstorming the title. We also thank Jens Tuyls for the helpful discussions.

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

Table 5: **Pretraining compute and setup for each Kraken configuration**

| Model | Layers | Embedding Dimension | Attention Heads | Total Params. | A100 GPU Hrs | Initial Learning Rate |
|---|---|---|---|---|---|---|
| Kraken 2-way | 12 | 678 | 12 | 124M | 480 | $2.5e-4$ |
| Kraken 4-way | 12 | 504 | 12 | 124.5M | 500 | $2.5e-4$ |
| Kraken 6-way | 12 | 418 | 12 | 123.2M | 800 | $2.5e-4$ |
| Kraken 2-way | 24 | 888 | 16 | 350M | 1000 | $1.5e-4$ |
| Kraken 4-way | 24 | 644 | 16 | 353.4M | 1750 | $1.5e-4$ |
| Kraken 4-way | 24 | 960 | 16 | 761M | 1800 | $1.5e-4$ |

# A Appendix

## A.1 Training setup and compute requirements

For all pretrained models presented in Section 4.1, we used a similarly sized GPT-3 (7) model's hyperparameters as the basis and followed the procedure outlined in Section 3.2 to calculate the embedding dimension. We did not make an effort to optimize the codebase used for training which builds off of nanoGPT (29). It is possible to replicate pretrained models by extending nanoGPT to implement the new forward pass as described in Algorithm 1. The Adam optimizer was used to train all models along with a cosine learning rate decay with linear warmup. Initial learning rates and the approximate GPU hours required to train each configuration are presented in Table 5. All models were trained for $300,000$ gradient steps. Only the $761M$ parameter model was trained on a node with 80GB A100 GPU machines. The other configurations were trained on 40GB A100 machines and consequently use many more gradient accumulation steps. This is why the largest model required a similar number of GPU hours as the next largest.

Weights for all pretrained GPT-2 models used when evaluating SuperGLUE performance (18) (45) (30) (59) (16) (5) (22) (6) (40) (46) (41) (33) in Section 4.2 were obtained from HuggingFace. Since the focus of this work is on illustrating the efficiency gains and language modeling capabilities, we do not implement any safeguards that will filter for biased and/or harmful content. Initial experiments that tried various variations of the model architecture required about another 1,000 hours of A100 compute.

## A.2 Overlap plugin implementation

We used TensorRT-LLM version *0.12.0.dev2024073000* throughout the evaluation. CUDA allows kernels to be launched on different streams and depending on resource availability, these kernels may be executed in parallel. However, TensorRT does not support multi-stream execution across plugins. We circumvented this limitation by implementing a Singleton that can manage a dedicated stream and global memory meant for launching collectives. This allows different instances of the plugin to launch and synchronize kernels on the same stream. Each instance of the plugin can either: trigger an AllReduce op on a separate low-priority CUDA stream or synchronize the stream to ensure that a previously launched AllReduce completes. The plugin also implements the functionality provided in the existing GEMM plugin. This allows us to perform the following within the Multi-Head Attention block:

1. The AllReduce is launched on a dedicated CUDA stream just before the GEMM used to compute the Query, Key, and Value matrices (GEMM $QKV$ 1) via the plugin

2. All intermediate compute is performed using existing kernels such as FlashAttention

3. The CUDA stream that the AllReduce op was placed on is synchronized after the GEMM $W_{out}$ 1, also via the Plugin

This approach allowed us to overlap the collective with all the computation required for Multi-Head Attention and is the only addition to TensorRT-LLM aside from the definition of Kraken. However, it also requires two extraneous memory copy operations that can be avoided if the library adds support for multi-stream execution.

Table 6: **Inference latency in milliseconds for 4-way parallelism**

| Model Size | Context Length | Standard | Parallel Attn. + FeedForward | Kraken |
|---|---|---|---|---|
| 1.3B | 128 | 3.7 | 3.3 | 3.3 |
| 1.3B | 2048 | 17.0 | 13.6 | 13.2 |
| 6.7B | 128 | 8.3 | 7.0 | 5.8 |
| 6.7B | 2048 | 48.2 | 42.1 | 38.0 |
| 13B | 128 | 13.0 | 11.1 | 10.7 |
| 13B | 2048 | 83.7 | 73.8 | 66.9 |
| 65B | 128 | 12.7 | 11.2 | 8.5 |
| 65B | 2048 | 84.6 | 79.5 | 63.8 |
| 175B | 128 | 24.6 | 22.5 | 19.9 |
| 175B | 2048 | 243.7 | 230.8 | 158.9 |

Table 7: **Inference latency in milliseconds for 8-way parallelism**

| Model Size | Context Length | Standard | Parallel Attn. + FeedForward | Kraken |
|---|---|---|---|---|
| 1.3B | 128 | 4.3 | 3.4 | 3.2 |
| 1.3B | 2048 | 15.9 | 11.7 | 9.7 |
| 6.7B | 128 | 7.1 | 5.7 | 4.7 |
| 6.7B | 2048 | 37.3 | 29.2 | 27.6 |
| 13B | 128 | 10.7 | 8.4 | 6.7 |
| 13B | 2048 | 58.8 | 46.9 | 42.6 |
| 65B | 128 | 8.7 | 7.2 | 6.2 |
| 65B | 2048 | 55.9 | 48.9 | 42.9 |
| 175B | 128 | 16.9 | 14.4 | 12.4 |
| 175B | 2048 | 125.1 | 114.7 | 98.0 |

## A.3 Time To First Token in milliseconds

Table 6 contains the results from Figure 4 but in terms of milliseconds. Similarly, Table 7 presents the results from Figure 5. For the 65B and 175B configurations, we ran into CUDA Out-of-Memory errors when running full sized engines because each device has only 40GB of memory. To avoid this, we reduce the number of layers to $\frac{1}{4}$ the original number. Nonetheless, the relative latency comparison should be unaffected because the runtimes of individual layers are identical.

