# OpenReview forum: "Kraken: Inherently Parallel Transformers For Efficient Multi-Device Inference"
_NeurIPS.cc/2024/Conference — NeurIPS 2024 poster_

### Official Review · Reviewer_6qbN · 2024-07-04

**Soundness:** 3
**Presentation:** 4
**Contribution:** 3
**Rating:** 6
**Confidence:** 4

**Summary:**

This paper proposes to modify standard transformer architectures to use less allreduce operations from tensor parallelism. Specifically, the proposal is to allow each shard of the hidden state to operate as an individual complete hidden state during attention layers and each shard of the attention layer to operate independently. Consequently, allreduce is only needed after FFN layers and can overlap in time with attention calculation. Experiments of up to 760M models show that the new architecture can achieve accuracy that is almost as good as same-sized GPT2. Latency projection suggests 1.5X speed up in context encoding for 175B-size models.

**Strengths:**

Please see the summary. The proposal is simple yet makes sense. The accuracy and latency results are promising.

**Weaknesses:**

The biggest weakness of this paper is that accuracy results are only available up to 760M model size. There remains big uncertainty for larger models where the proposal matters. Also there seem to be consistently weak results on the ReCoRD task, is there an explanation on why?

Another weakness of this paper is that the model training is committed to a specific tensor-parallelism degree. This is not a huge issue in practice but still a weakness.

**Questions:**

Please see above.

**Limitations:**

Yes

---

> ### Author Rebuttal · Authors · 2024-08-07
>
> **Performance on ReCoRD:**
>
> Please see the common response. This was due to a bug in the LM Evaluation Harness (Issue 1647 on GitHub).
>
> **Results only up to 760M parameters:**
>
> Indeed, it would have been great to evaluate models that are several billion parameters large. Unfortunately, we do not have access to the amount of compute necessary for such an endeavor. Nonetheless, we hope the results presented here will encourage others to try this approach given how prevalent and expensive Transformer inference can be.

---

> > ### Comment · Reviewer_6qbN · 2024-08-12
> > **Thanks for the response**
> >
> > Thanks for the response. My ratings remain the same.

---

### Official Review · Reviewer_vwQM · 2024-07-09

**Soundness:** 2
**Presentation:** 3
**Contribution:** 3
**Rating:** 6
**Confidence:** 4

**Summary:**

The paper introduces a modification to the standard Transformer architecture aimed at reducing inter-device communication during inference in multi-device systems. By predetermining the degree of model parallelism, computations on each device can operate independently, allowing collective operators to overlap with compute tasks. Evaluations conducted using the SuperGLUE benchmark suite demonstrate that the proposed models maintain similar perplexity levels compared to standard Transformers while achieving notable speed improvements.

**Strengths:**

* **Novelty and contribution**: The paper presents a novel approach to parallelizing the Multi-Head Attention (MHA) and Feed-Forward Network (FFN) layers, allowing them to run concurrently with the AllReduce operations. This innovative strategy shows promise and is orthogonal to existing methods like FlashAttention and Speculative Decoding.
* **Soundness**: The design and methodology, while simple, are robust and well-justified. The authors provide clear explanations and solid rationales for their approach, enhancing the credibility of their work.
* **Clarity**: The paper is well-structured and written in a clear, concise manner, making it easy to follow and understand.

**Weaknesses:**

* **Limited benchmark**: The evaluation benchmarks primarily focus on language tasks, neglecting the application of Transformers to vision tasks.
* **Limited comparative analysis**: The paper lacks comparisons with other existing baselines, including ablation studies and sensitivity analyses. While the authors compare their approach with models running Feed-Forward Networks (FFN) in parallel with Attention, they do not explore sensitivity to varying inter-device bandwidths or different compute-to-communication ratios.

**Questions:**

* Could there be additional evaluations on a more diverse set of benchmarks? Can the proposed approach be extended to applications in vision-based tasks?
* How does the approach benefit systems with varying inter-device configurations and bandwidth capacities?

**Limitations:**

The authors have thoroughly discussed the limitations of their approach. From the checklist, it appears that the approach does not pose significant negative societal implications, apart from the associated costs and resources required for training large models.

---

> ### Author Rebuttal · Authors · 2024-08-07
>
> **Applicability to vision tasks:**
>
> Yes, our approach will also translate to encoder-only Transformers including those used in vision tasks such as ViT[4]. Nonetheless, we narrowed the scope of this work to focus our resources on the decoder-only Transformer models that are typically the largest and consequently the most expensive to run inference on.
>
> 4: “Image is worth 16x16 words…”, Dosovitskiy et al. 2020
>
>
> **Varying device interconnects and compute/communication ratios:**
>
> We do present results at various compute/communication ratios because the cost of the AllReduce scales linearly with sequence length (and embedding size) but Attention itself is quadratic (Figure 4). For longer sequences, communication will occupy a smaller fraction of overall runtime.
>
> Additionally, our evaluation platform (NVSwitch) offers the most performant interconnect between A100 GPUs that we are aware of. As such, assuming that the compute capability of each device is unchanged, the presented latency improvements are a lower bound when compared to alternative topologies where some connections traverse PCIe and consequently suffer from limited bandwidth.

---

> > ### Comment · Reviewer_vwQM · 2024-08-12
> >
> > Thank you to the authors for the response. I will maintain my current rating.

---

### Official Review · Reviewer_ptrK · 2024-07-12

**Soundness:** 3
**Presentation:** 3
**Contribution:** 3
**Rating:** 5
**Confidence:** 4

**Summary:**

This paper propose Kraken, a new evolution of the standard Transformer to reduce the communication cost of inference. Kraken overlaps the collective operations with computing, therefore achieves smaller pre-filling time cost. Kraken is specially designed for tensor parallelism on multi-device environments.

**Strengths:**

1. The focus on accelerating the pre-filling stage of inference by reducing communication overhead addresses a significant and challenging issue that merits further research.
2. Kraken preserves much of the standard Transformer architecture while effectively reducing communication overhead.
3. The solution of Kraken is is straightforward yet impactful.
4. Kraken maintains the model’s accuracy despite its optimizations.

**Weaknesses:**

1. The paper lacks a detailed examination of related works that also focus on reducing communication costs. Overlapping techniques to accelerate inference have been widely explored in both industrial and academic contexts, as demonstrated by works such as [1, 2] and tools like NCCL. A clearer differentiation between Kraken and these existing solutions is necessary.

> [1] Wang, Shibo, et al. "Overlap communication with dependent computation via decomposition in large deep learning models." (ASPLOS’22)

> [2] Rajbhandari, Samyam, et al. "Zero-infinity: Breaking the gpu memory wall for extreme scale deep learning." (SC’21)

2. Sharing token and positional embeddings across all sub-layers might introduce limitations in how effectively the model can learn positional nuances across different parts of the input sequence. Thus, except the evaluation  in 4.1 and 4.2, the paper should clarify why the new architecture does not affect the accuracy of the model.

**Questions:**

This paper focuses on an important issue and proposes an simple yet effective solution. However, it has several limitations that prevent the paper from being accepted.

Firstly, a comprehensive study about related works is lacked. It is hard for me to get the key idea of this paper. What is the advantages of Kraken compared with other works?

Secondly, although the paper has preserved most of the standard Transformer architecture, it still requires to re-train a model from scratch, which is extremely costly, especially for LLMs. The necessity of altering the Transformer architecture should be clarified. Specifically, it should be determined whether Kraken could be designed as a plug-in for any Transformer-based model and can work directly or require minimal re-training.

Besides, as tools like DeepSpeed, PagedAttention, and Flash Attention are widely adopted in current LLM inference, it is critical that Kraken demonstrates compatibility with these tools to enhance performance and practicality.

**Limitations:**

Please refer to the Weaknesses and Questions.

---

> ### Author Rebuttal · Authors · 2024-08-07
>
> **Comparison to Related Work:**
> Our approach is readily compatible with nearly all related work such as DeepSpeed, FlashAttention, and PagedAttention. Our evaluation (Figure 4) uses FlashAttention and TensorRT-LLM’s implementation of AllReduce which is even more performant than NCCL on systems equipped with NVLink.
>
> The fundamental contribution is to reformulate the Transformer layer and remove the true data-dependency introduced by the collectives necessary for tensor parallelism. This is illustrated by Figure 3. The closest point of comparison would be models that use parallel Attention and FeedForward which we do evaluate (Figure 4). Our model-level change is compatible with most system-level optimizations for Transformer training and inference.
>
> By virtue of the architecture, we don’t require the techniques proposed by Wang et al.[1] unless a Kraken model is parallelized on more devices than its degree of parallelism.
>
> **Pre-training and PlugIn:**
> While Kraken requires training from scratch, any plug-in we develop would still need to account for the true data dependency between layer activations in the original architecture. This imposes a hard limit on efficiency. In fact, the suggested plug-in would mostly take the form of the approach proposed by Wang et al. [1]
>
> **Sharing token and positional embeddings:**
> The absence of shared positional and token embeddings would complicate weight tying[3] given that we are computing only one probability distribution at the last layer.
>
> 3: “Using the output embedding…”, Press et al. 2017

---

> > ### Comment · Reviewer_ptrK · 2024-08-13
> >
> > Thank you for the explanation. I now clearly understand Kraken's key idea and how it differs from related works such as [1, 2]. It's a notable strength that Kraken is compatible with many existing tools like TensorRT. However, the requirement to train from scratch significantly impacts its application in LLMs due to the high cost of re-training. Considering both its strengths and weaknesses, I’d like to adjust my rating to 5.

---

> > > ### Author Response · Authors · 2024-08-13
> > > **Energy Usage and Infrastructure Allocation For Training vs Inference**
> > >
> > > We thank the reviewer for appreciating how our approach is compatible with existing techniques. Yes, training from scratch is indeed very expensive. However, as the usage of Transformers grows, inference will require a much larger fraction of the available power and infrastructure. As an example, take the case study presented by Wu et al. [1]; here, the authors find that 70% of available power at a major tech company goes to ML inference (Figure 3) as opposed to 20% for training. Moreover, this work[1] and data is from 2022 which is a little before the current "Gen AI" trend that has taken over the industry.
> > >
> > > Sustainable AI: Environmental Implications, Challenges, and Opportunities, Wu et al., MLSYS 2022.

---

### Author Rebuttal · Authors · 2024-08-07

We thank all the reviewers for reading our work, providing helpful feedback, and finding promise in our approach.

After submission, we learned of bugs in LM Evaluation Harness that affected the scores on some SuperGLUE benchmarks such as ReCoRD. We have evaluated all models again using the most recent version of the toolkit; **updated results for Tables 2 and 3 are available in the attached PDF**.

---

### Decision · Program_Chairs · 2024-09-25

**Decision:**

Accept (poster)

**Comment:**

This paper solved an important problem: improving the parallel efficiency for generative inference. An interesting modification is made to the standard Transformer architecture aimed at reducing inter-device communication during inference in multi-device systems. The solution is solid and has good novelty, and the experimental section is well-designed to verify the design of the proposed solution.